

# Changes in long non-coding RNA transcriptomic profiles after ischemia-reperfusion injury in rat spinal cord

Zhibin Zhou[1,*], Bin Han[1,*], Hai Jin[2], Aimin Chen[1] and Lei Zhu[1]

[1] Department of Orthopaedics, Changzheng Hospital, Second Medical University, Shanghai, China
[2] Department of Neurosurgery, 202 Hospital of China Medical University, Shengyang, Liaoning, China
[*] These authors contributed equally to this work.

## ABSTRACT

With the aim of exploring expression profiles and biological functions of long non-coding RNA (lncRNA) and mRNAs after spinal cord ischemia-reperfusion injury (SCII), differentially expressed lncRNAs (DElncRNAs) and mRNAs (DEmRNAs) in rat spinal cords were identified following SCII through high-throughput RNA sequencing. In total, 1,455 lncRNAs and 6,707 mRNAs were observed to be differentially expressed ($|\text{Fold Change}| \geq 2$ and $P < 0.05$) after SCII, including 761 up-regulated and 694 down-regulated lncRNAs, 3,772 up-regulated and 2,935 down-regulated mRNAs. Gene ontology and KEGG pathway analysis showed that the DElncRNAs and DEmRNAs were implicated in many different biological processes and pathways. Further, lncRNA-mRNA co-expression networks were built to explore the potential roles of these DElncRNAs. Our results demonstrate genome-wide lncRNA and mRNA expression patterns in spinal cords after SCII, which may play vital roles in post-SCII pathophysiological processes. These findings are important for future functional research on the lncRNAs involved in SCII and might be critical for providing new insight into identification of potential targets for SCII therapy.

# INTRODUCTION

Spinal cord ischemia-reperfusion injury (SCII) is a serious complication following surgeries that implicates the descending and thoracoabdominal aorta, and can lead to major impairments to functioning of the bowel, bladder, sex and lower extremities (*Wan et al., 2001*). Although great efforts have been made to develop medical therapeutics for SCII, no efficacious pharmacological or surgical intervention is currently available in clinic to attenuate this injury (*Xu et al., 2014*; *Zhu et al., 2015*). Thus, a better mechanistic understanding of SCII is still needed for developing neuroprotective strategies to this devastating complication.

The pathological process of SCII includes two phases: the immediate phase is related to acute ischemia, while the delayed phase involves both ischemic cellular death and reperfusion injury (*Zhu et al., 2013*). The immediate phase cannot be changed, but injuries

Corresponding authors
Aimin Chen,
aiminchen@smmu.edu.cn
Lei Zhu, hailangzhulei@smmu.edu.cn

caused in the delayed phase can possibly be mitigated by effective therapy. Numerous studies have shown that apoptosis, amino acid toxicity, and calcium overload induced by ischemia-reperfusion injury contribute to neuronal cell death in the pathological processes of SCII (*Fan et al., 2011*; *Ni, Cao & Liu, 2013*), however, the pathophysiology of SCII varies from time to time. For example, cytokines, which have been proven to play critical role in the delayed phase of SCII, were previously reported to peak in the serum at the early stage of reperfusion (within the first 24 h), while studies further found that most of them actually showed a biphasic response (peaked at 6 and 36 to 48 h) (*Smith et al., 2012*). Besides that, current evidence also suggests that the spinal cord tissue at the late stage (48 h) following SCII showed severer and more typical histological feature, which consistent with the degree of functional impairment of the hind limb (*Hasturk et al., 2009*; *Smith et al., 2012*). Therefore, revealing the changes in pathophysiology of SCII at late stage of reperfusion in the delayed phase might be of great help in understanding the biological mechanism of SCII.

Long non-coding RNA (lncRNA) represents a class of transcripts longer than 200 nucleotides, but with no coding function (*Mercer, Dinger & Mattick, 2009*). Accumulating evidence has shown that lncRNAs have various biological functions in RNA processing, transcription and translation via acting as important gene expression regulators (*Cech & Steitz, 2014*). Numerous reports have highlighted that aberrant expression of lncRNAs may be crucial in physiological and pathological processes of various human diseases, including myocardial infarction (*Wu et al., 2018*), osteoarthritis (*Liu et al., 2014*), diabetic nephropathy (*Wang et al., 2018a*) and cancers (*Liu, Xie & Zhang, 2018*; *Wang et al., 2018b*; *Zheng et al., 2018*). In recent years, studies on the function of lncRNA in SCII have also gradually increased. Liu et al. proved that hydrogen sulfide up-regulated lncRNA CasC7 and reduced neuronal cell apoptosis in SCII (*Liu et al., 2018*), Wang et al. revealed that lncRNA MALAT1 played a neuroprotective role in a rat SCII via regulating miR-204 (*Qiao et al., 2018*), and a resesrch group indentified expression profile of lncRNAs in the spinal cord at the early stage of reperfusion (2 h post-cardiac I/R) by high-throughput RNA sequencing (*Wang et al., 2019*). However, lncRNA expression pattern and its roles in SCII at the late stage of reperfusion have not been reported yet.

Here, to further investigate the pontial roles of lncRNA in SCII, we screened genome-wide expression patterns of lncRNAs and mRNAs in spinal cords from SCII and control rats via RNA sequencing. Several differentially expressed lncRNAs (DElncRNAs) and mRNAs (DEmRNAs) were validated through quantitative reverse transcription polymerase chain reaction (qRT-PCR). In addition, we performed gene ontology (GO) and KEGG pathway analyses and built co-expression networks to explore the functions of DElncRNAs and DEmRNAs. These results provide a basis for future functional research on the lncRNAs involved in SCII and might be critical for providing new insights into identification of potential targets for SCII therapy.

## MATERIALS & METHODS

### Spinal cord ischemia reperfusion injury model establishment

Male Sprague-Dawley rats (weight, 200–250 g) were purchased from the Animal center of the Second Military Medical University (Shanghai, China). All animal experiments were performed according to the guidelines of the Animal Ethics Committee of the Second Military Medical University (Shanghai, China). Thirty rats were randomly divided into the SCII group and sham group (nine in each). In SCII group, the SCII model was established as previously described (*Li et al., 2014*). After identification of the aortic arch branches, ischemia was achieved by cross-clamping the aortic arch for 14 min. Obstruction was confirmed by a 90% blood flow decrease in the tail artery using a Laser Doppler Bloodflow Imager (Perimed, Stockholm, Sweden). The clamps were released to reperfuse for 48 h. Rats in sham group that underwent the same procedure without occlusion were used as controls.

### RNA extraction and high-throughput RNA sequencing

Spinal cord segments between L4 and L6 were harvested 48 h after reperfusion. In total, six samples (three from the SCII group and three from the sham group) were subjected for high-throughput RNA sequencing. For each sample, three random rats spinal cord tissues in the same group were mixed into one sample before RNA extraction. Total RNA was extracted using the mirVana miRNA Isolation Kit (Ambion, Texas, USA) following the manufacturer's protocol. RNA integrity was evaluated using the Agilent 2100 Bioanalyzer (Agilent, California, USA). The samples with RNA Integrity Number (RIN) ≥ 7 were subjected to the subsequent analysis. The libraries were constructed using TruSeq Stranded Total RNA with Ribo-Zero Gold (Illumina, California, USA) according to the manufacturer's instructions. After validating the size and purity by Agilent Technologies 2100 Bioanalyze (Agilent, California, USA), these libraries were sequenced on the Illumina HiSeqTM 4000 sequencing platform (Illumina, California, USA) and 150 bp paired-end reads were generated. Libraries construction and RNA-sequencing were performed by OE Biotech (Shanghai, China).

### Bioinformatics analysis

We obtained about 99 million raw RNA-sequencing reads from each sample. First, raw RNA-Seq data were flattened by removing adapter sequences and filtering out low quality reads using Trimmomatic software (*Bolger, Lohse & Usadel, 2014*). The clean reads were obtained by quality detection of the obtained reads through fastqc software (www.bioinformatics.babraham.ac.uk/projects/fastqc/), and then they were mapped to the rat reference genome (Rattus_norvegicus.Rnor_6.0) using hisat2 (*Kim, Langmead & Salzberg, 2015*). The transcript abundances of lncRNA and mRNA were quantified by normalized expression values as FPKM (Fragments Per kb Per Million Reads) using bowtie2 (*Langmead & Salzberg, 2012*) and eXpress (*Roberts & Pachter, 2013*). To identify differentially expressed lncRNAs or mRNAs between the spinal cord after ischemia reperfusion injury and the controls, differential expression analyses were performed

**Table 1  qRT-PCR primer sequences.**

|  | Forward primer(5′->3′) | Reverse primer (5′->3′) |
|---|---|---|
| TCONS_00018593 | TAAGGCACTCTGGGGACGAA | GGGATTAGATGCTGTGGGGC |
| TCONS_00018687 | GGACATCCCAAAGATGGCGT | GGCAGGAGCAGGTTCTTAGT |
| NONRATT013040.2 | GGCGCCTGTGAGTAGATGAA | CCGACAAAATTCGGCTCGTG |
| NONRATT013069.2 | ATCCACCGACAGTTGGAACC | GTACGCAGAATGACCTCGCT |
| NONRATT007222.2 | AGTCAAGTTCAAGCCGTCCC | CCACTAGGAGTGACCTGTGC |
| TCONS_00008647 | AGAGTCAAGAGCGGACCCA | GTCCCAAACGTGTTTGTCCT |
| NONRATT019127.2 | CTCCGTTCCGGGCTCTAGT | AACGGGGAGACAGGATGCTT |
| TCONS_00020362 | GAAAGCTGGCCTGGGTTCTA | GCTCAGGACCCTAACCTGTG |
| NM_001191578.1 | CCATGGTGACTGAGAGTCCG | CTGTCTCCAGGGACAAGCAG |
| XM_017592225.1 | GAGGCTGGAGGAAGGAGAGA | ATGTTGACGTCTGGGGTGTC |
| XM_017592198.1 | CCATGGTGACTGAGAGTCCG | CTGTCTCCAGGGACAAGCAG |
| NM_001191577.1 | CTGTGAAAAGCTGGCAACCC | TCAGGCTTCCAAACGGAGTC |
| NM_017104.2 | GTCGACAACTTTGCCACCAC | GGTACCACCACAGAATGGGG |
| NM_013037.1 | GTCAGTGCACAAAAGGCTGG | ATTTCCGCTTGGGGGCATAA |
| NM_001142366.2 | TAGCCGGTCTCCTAGCAGTT | CTTTGCCTTGCAGGTTGGTC |
| NM_001191043.1 | CACTACACTGTGAGGCCCAG | GGCTGCCTCACACTAACCAT |

using DESeq package. The $p$-value <0.05 and |fold change|$\geq$ 2 were selected as the criteria for significantly DElncRNA or DEmRNA.

## Quantitative real-time PCR (qRT-PCR)

To validate the reliability of the RNA sequencing results, We randomly selected and analyzed 8 DElncRNAs and 8 DEmRNAs by qRT-PCR analysis. L4–L6 spinal cord tissues were collected from the SCII and sham groups ($n = 6$ per group). Total RNA was extracted using the TRIzol reagent (Invitrogen, NY, USA) and reversed transcribed into cDNA using random primers. Quantification of lncRNA and mRNA was performed using an ABI PRISM® 7500 Sequence Detection System (Applied Biosystems, Forster, CA, USA). Relative lncRNA expression was normalized to the U6 expression and the expression of mRNA level was normalized to GAPDH using the 2-$\Delta\Delta$CT method. Three independent experiments were conducted for each sample. The primer sequences are shown in Table 1.

## Functional and pathway enrichment analyses

GO and KEGG pathway analyses were conducted to predict the potential functions of DEmRNAs and DElncRNAs based on their co-expressed DEmRNAs. GO analysis was performed to explore potential functions of genes and gene products from three aspects: biological process (BP), cellular component (CC), and molecular function (MF). KEGG pathway analysis was conducted to predict the involvement of the differentially expressed genes in the biological pathways. The top 10 enriched GO terms and top 20 enriched pathways were ranked by enrichment score ($-\log10$ ($p$-value)) identified by the Database for Annotation, Visualization, and Integrated Discovery (DAVID; https://david.ncifcrf.gov/).

**Table 2** The results of RNA-Sequencing and clean reads mapping to the reference genome for each group.

| Sample | raw_reads | clean_reads | Total reads | Total mapped reads |
|---|---|---|---|---|
| SCII1 | 98.19M | 95.49M | 95487232 | 93482612(97.90%) |
| SCII2 | 99.73M | 96.94M | 96938560 | 95121138(98.13%) |
| SCII3 | 99.55M | 96.77M | 96773454 | 94651018(97.81%) |
| sham1 | 99.03M | 95.94M | 95938648 | 94021084(98.00%) |
| sham2 | 98.18M | 95.22M | 95220464 | 93256257(97.94%) |
| sham3 | 99.20M | 96.43M | 96426788 | 94543525(98.05%) |

## Co-expression of lncRNAs/mRNAs and functional prediction

The functions of lncRNAs were predicted by annotating the function of the co-expressed mRNAs. A co-expression network was built to explore the interaction between DElncRNAs and DEmRNAs in SCII pathogenesis. The Pearson's correlation coefficient (PCC) was calculated between the expression levels of each DElncRNA-DEmRNA. We then selected co-expression DElncRNA-DEmRNA pairs with value of |PCC| ≥0.80 and $p$-value <0.05 for network construction (*Alkan et al., 2017*).

Genomic localizations of the paired lncRNAs and mRNAs were identified for cis prediction. The co-expression nearby gene, which is less than 100 kb upstream or downstream from the lncRNA, can act as the potential target regulated by the lncRNA in a cis manner, while a trans-regulator is one that does not meet this criterion. The RIsearch-2.0 software (*Alkan et al., 2017*) was used to identify target genes in trans, with the parameter set as the base number of direct interactions between lncRNA and mRNA ≥10 and free energy ≤ −50.

## RESULTS

### High throughput RNA sequencing and genome-wide read mapping

L4 to L6 spinal cord segments were harvested for total RNA isolation and Illumina TruSeq library construction (Data available at GEO: GSE138966). 95.49 M, 96.94 M and 96.77 M high quality sequence reads were obtained from three samples of the SCII group, and 95.94 M, 95.22 M and 96.43 M clean reads were generated from three control tissues of the sham-operated group (Table 2). All clean reads were mapped to the rat reference genome using hisat2. Nearly 98% of the reads mapped to the rat genome; 75.25%–81.13% of the reads mapped to unique genomic regions among the aligned fragments (Table 2), indicating the reliability of the data. In total, 25,328 lncRNAs and 55,760 mRNAs were identified.

### The expression profile of lncRNA and mRNA in the SCII model

In this study, 1,455 lncRNAs and 6,707 mRNAs were significantly differentially expressed, with |fold change|≥2.0, $P < 0.05$ and FDR <0.05. In total, there were 761 up-regulated lncRNAs, 694 down-regulated lncRNAs, 3,772 up-regulated mRNAs, and 2,935 down-regulated mRNAs. Scatter plots analyses showed the expression signatures (Figs. 1A and 1C). Hierarchical clustering expression showed significant differences in the spinal cord
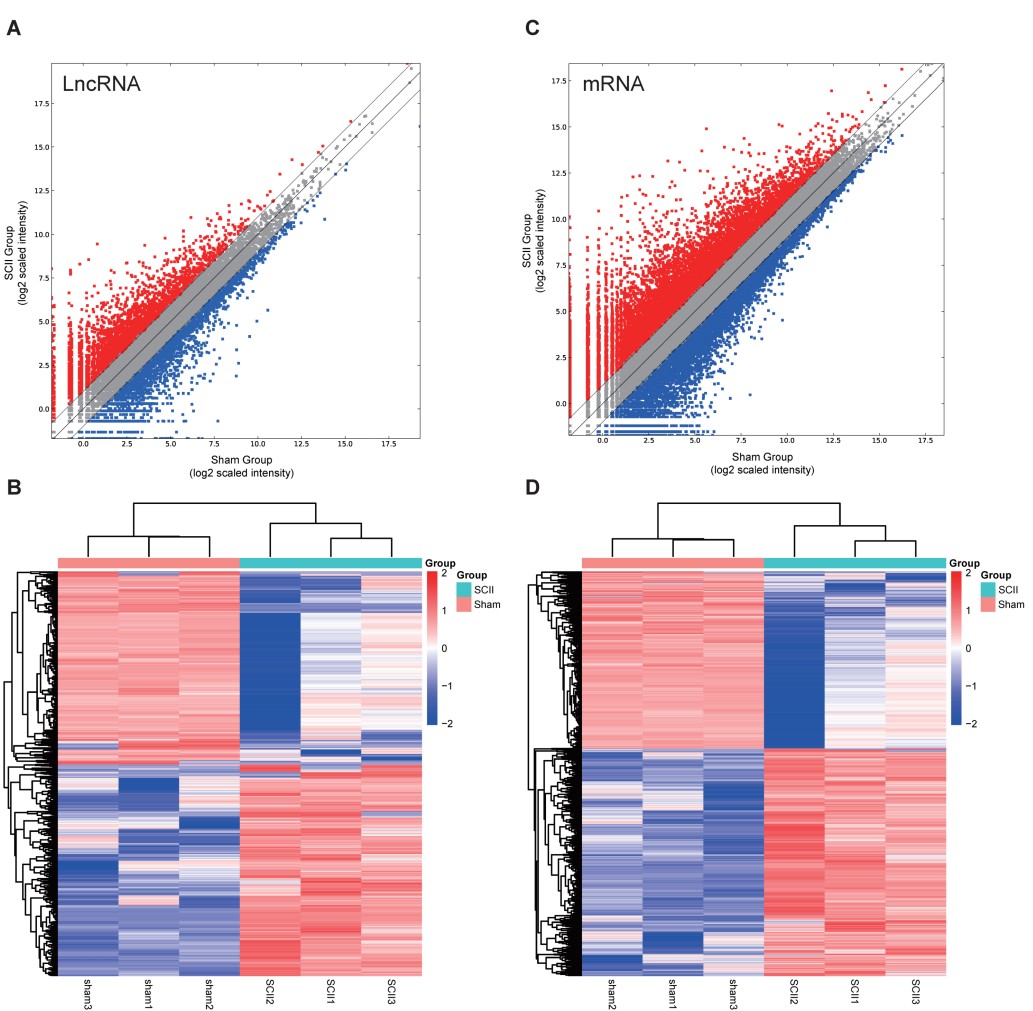

**Figure 1** **Expression profiles of LncRNAs and mRNAs in rat spinal cord after ischemia-reperfusion injury.** Scatter-plot for comparing global expression profiles of lncRNAs (A) and mRNAs (C) in the spinal cord between the SCII and sham-operated rats. X and Y axes indicate the mean normalized signal values (log2 scaled). Red points represent up-regulated lncRNAs or mRNAs while blue points represent down-regulated lncRNAs or mRNAs. The heat map shows hierarchical clustering of DElncRNAs (B) and DEmR-NAs (D) in the spinal cord between the SCII and sham-operated rats. The color scale indicates the expression of DElncRNAs and DEmRNAs. Red and blue indicates up- and down-regulation, respectively.

between SCII rats and control rats (Figs. 1B and 1D). Among the top 10 DElncRNAs, NONRATT007222.2 and NONRATT019127.2 were the most significantly up-regulated and down-regulated in the spinal cord after SCII (Table 3). Moreover, Csf3 and Ankrd9 were the most significantly up- and down-regulated mRNAs in spinal cord after SCII (Table 4). Further, transcripts were proved to be distributed on all chromosomes (Fig. 2).

## Validation of lncRNAs and mRNAs

Eight mRNAs and 8 lncRNAs were randomly selected to verify the high throughput RNA sequencing results in six sample pairs by qRT-PCR. We found that the

**Table 3    Top 10 up-regulated and down-regulated mRNAs in SCII.**

| ID | Gene | Log2FC | P_value |
|---|---|---|---|
| **Up-regulation** | | | |
| NM_017104.2 | Csf3 | 11.93599648 | 5.20E-39 |
| NM_013037.1 | Il1rl1 | 11.39596538 | 0.001080239 |
| XM_006247931.3 | Slc16a3 | 11.28927512 | 0.000375611 |
| NM_203493.3 | Dmp1 | 10.68101399 | 0.02825727 |
| XM_008766983.2 | Il1r2 | 10.65398384 | 6.52E-06 |
| NM_012589.2 | Il6 | 10.64190268 | 3.75E-58 |
| NM_053647.1 | Cxcl2 | 10.34386563 | 8.60E-39 |
| XM_006246949.3 | LOC497963 | 10.14662547 | 9.86E-07 |
| NM_053822.2 | S100a8 | 9.93922593 | 0.000520228 |
| NM_138522.1 | Cxcl3 | 9.859847615 | 8.34E-20 |
| **Down-regulation** | | | |
| NM_001012112.1 | Ankrd9 | −7.681552228 | 0.003776415 |
| XM_017600715.1 | LOC102553010 | −7.508339301 | 1.06E-13 |
| NM_012705.1 | Cd4 | −7.205888701 | 0.02515971 |
| XM_017599444.1 | Aasdh | −7.136997274 | 0.000145736 |
| NM_001271135.1 | Ulk3 | −6.983610521 | 0.002151312 |
| XM_017592918.1 | Crot | −6.929796154 | 0.000487074 |
| XM_017602365.1 | Zcchc16 | −6.837049544 | 0.000795313 |
| XM_008760819.2 | Cdh18 | −6.687559573 | 0.000603272 |
| XM_017594898.1 | Slc4a8 | −6.631820122 | 0.009401042 |
| XM_017596528.1 | Speg | −6.425796396 | 0.001340429 |

expressions of lncRNAs NONRATT007222.2, NONRATT013040.2, TCONS_00018593 and TCONS_00008647 were up-regulated, while NONRATT019127.2, NONRATT013069.2, TCONS_00018687 and TCONS_00020362 were down-regulated. Of the eight selected mRNAs, NM_017104.2, NM_001191577.1, XM_017592198.1 and NM_013037.1 were up-regulated, and NM_001142366.2, NM_001191578.1, XM_017592225.1 and NM_001191043.1 were down-regulated in the SCII model (Fig. 3). The qRT-PCR results were in line with the high throughput RNA sequencing. Therefore, the results validated our high throughput RNA sequencing profile as highly reliable and showed that these lncRNAs and mRNAs might be involved in SCII pathogenesis.

## Gene ontology and KEGG Pathway annotation for differentially expressed genes

In GO and KEGG pathway enrichment analyses of DEmRNAs, we found 6707 mRNAs that were differentially expressed. GO enrichment analysis showed that the enriched GO biological processes for up-regulated genes in the SCII group were negative regulation of CD40 signaling pathway, positive regulation of T-helper 17 cell differentiation, cellular response to glucoside, cellular response to diterpene and cellular response to cyanide. The enriched GO cellular components for up-regulated genes in the SCII group were for the CRLF-CLCF1 complex, inner dense plaque of desmosome, outer dense plaque

**Table 4  Top 10 up-regulated and down-regulated lncRNAs in SCII.**

| ID | Locus | Length | Log2FC | P_value |
|---|---|---|---|---|
| **Up-regulation** | | | | |
| NONRATT007222.2 | chr12:2213723:2217002:+ | 861 | 9.8391916 | 1.37E-07 |
| TCONS_00008647 | Chr13:70748288:70749443:+ | 597 | 8.6841803 | 2.20E-20 |
| NONRATT009530.2 | chr14:84501194:84502293:+ | 1099 | 8.6388028 | 1.12E-11 |
| NONRATT027814.2 | chr8:71937230:71941941:+ | 784 | 8.5135019 | 0.03536145 |
| NONRATT004368.2 | chr10:47983254:47984472:+ | 1218 | 8.485743 | 2.71E-05 |
| NONRATT015075.2 | chr2:56892628:56893363:+ | 735 | 8.4529202 | 2.97E-08 |
| NONRATT001432.2 | chr1:220835102:220836600:+ | 1498 | 8.2443526 | 1.07E-07 |
| NONRATT025081.2 | chr6:92136234:92136940:- | 706 | 8.1274884 | 0.00038525 |
| NONRATT024498.2 | chr6:108087671:108090182:- | 2511 | 7.6101354 | 0.0012159 |
| TCONS_00026900 | Chr7:122665947:122710909:+ | 376 | 7.5871507 | 0.02229607 |
| **Down-regulation** | | | | |
| NONRATT019127.2 | chr3:8678461:8690286:- | 411 | −8.408993 | 7.61E-18 |
| NONRATT006212.2 | chr10:97733700:97735485:- | 1785 | −8.300526 | 0.0040744 |
| NONRATT026560.2 | chr7:14222096:14228950:- | 3792 | −7.64609 | 0.0021739 |
| NONRATT005960.2 | chr10:85655223:85658023:- | 2800 | −6.878412 | 4.03E-12 |
| NONRATT027045.2 | chr7:117180655:117187427:- | 959 | −6.779899 | 0.00030453 |
| NONRATT007881.2 | chr12:24543744:24548614:- | 1353 | −6.66036 | 0.04019041 |
| NONRATT015918.2 | chr2:1791288:1791544:- | 256 | −6.650513 | 0.00339351 |
| NONRATT030119.2 | chr9:73614969:73615827:+ | 280 | −6.580291 | 0.00026021 |
| NONRATT004679.2 | chr10:71858906:71860354:+ | 440 | −6.467266 | 0.01227824 |
| NONRATT023755.2 | chr5:156811469:156812092:- | 623 | −6.430205 | 0.00166601 |

of desmosome, integrin alphaL-beta2 complex and integrin alpha1-beta1 complex. The enriched GO molecular functions for up-regulated genes in the SCII group were chemokine (C-C motif) ligand 5 binding and NEDD8 ligase activity (Fig. 4A). Down-regulated mRNAs were enriched in dendrite arborization, maintenance of synapse structure, neuron projection, axon, neuronal cell body, extracellularly glycine-gated ion channel activity, protein binding and calcium ion binding of GO biological process, cellular component and molecular function (Fig. 4B).

Similarly, differentially expressed genes were analyzed using KEGG. We found that up-regulated genes in the spinal cord after SCII were involved in cytokine-cytokine receptor interaction (rno04060), TNF (rno04668) and NF-κB signaling pathway (rno04064) (Fig. 5A), while down-regulated genes were involved in axon guidance (rno04360), calcium (rno04020) and cAMP signaling pathway (rno04024) (Fig. 5B).

## lncRNAs/mRNAs co-expression analysis and functional prediction

Co-expression analysis was performed to understand the roles of lncRNAs according to their co-expressed mRNAs. 385 DEmRNAs and 221 DElncRNAs, which were composed of 606 nodes and 500 edges (Fig. S1) were involved. Additionally, positively co-expressed and negatively co-expressed networks were constructed to show two distinct co-expression

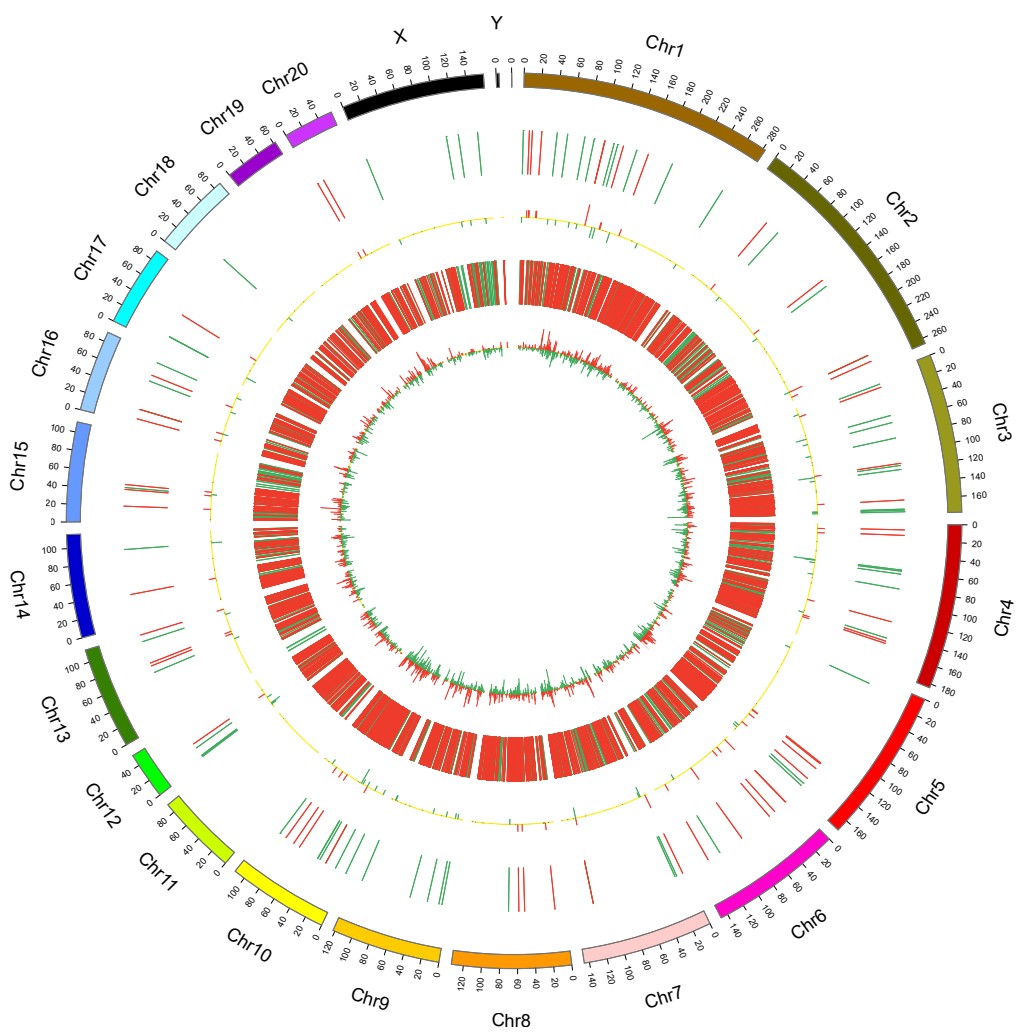

**Figure 2** **Circos plots representing the distribution of DElncRNAs and DEmRNAs on rat chromosomes.** The outermost layer of the circos plot is the chromosome map of the rat genome. The largest and larger inner circles represent all DElncRNAs detected by RNA-sequencing with fold change ≥2.0, $p < 0.05$, and FDR < 0.05. The increased or decreased lncRNAs are marked with red or green bars, respectively, and bar heights in the larger inner circle indicate numbers of DElncRNAs. The smaller and smallest inner circles represent all DEmRNAs detected by RNA-sequencing with fold change ≥2.0, $p < 0.05$ and FDR < 0.05. Increased or decreased mRNAs are marked with red or green bars, respectively, and bar heights in the smallest inner circle indicate numbers of DEmRNAs.

patterns of lncRNA-mRNA associated with SCII (Figs. 6A and 6B). Among these co-expressed mRNAs, the top three up-regulated DEmRNAs were Ermap, Hmox1and Il1r1, and these mRNAs were involved in regulation of cytokine production, apoptotic process, wound healing involved in inflammatory response, regulation of inflammatory response and positive regulation of neuron migration. The down-regulated DEmRNAs were Tmem255b, Smarcal1and Cpne5. These genes are associated with many functions, such as cellular response to DNA damage stimulus, neuron projection and regulation of dendrite extension. To better predict the functions of DElncRNAs identified in our study,

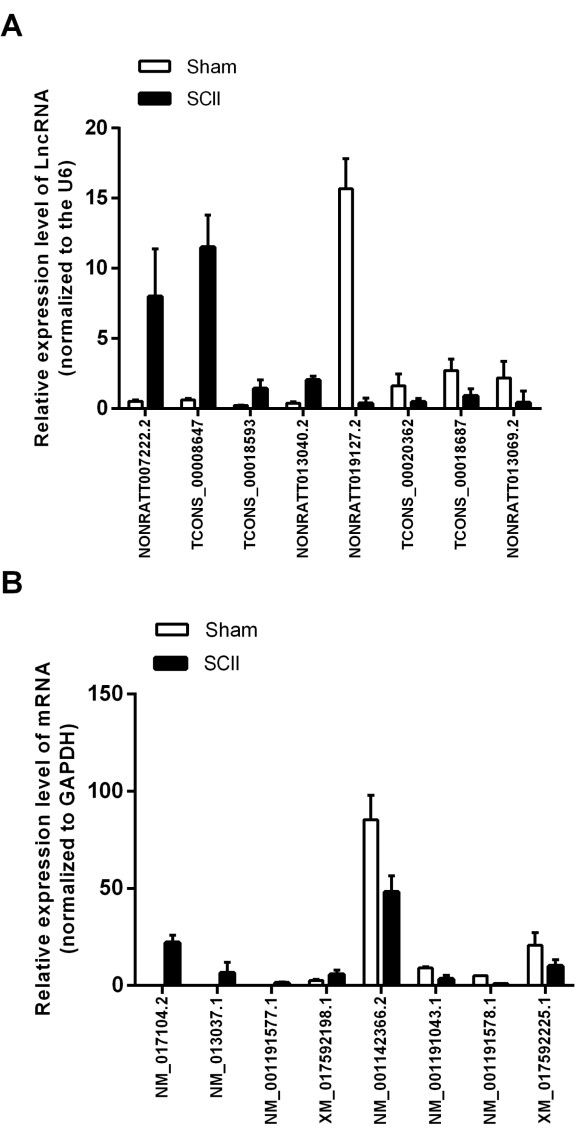

**Figure 3** **qRT-PCR validation of DElncRNAs and DEmRNAs in the spinal cord of SCII rats compared with matched tissues of sham-operated rats.** (A) Expression level of DElncRNAs determined by qRT-PCR (normalized to U6); (B) Expression level of DEmRNAs determined by qRT-PCR (normalized to GAPDH).

functional annotation of their co-expressed mRNAs were performed (Figs. 7A and 7B). The KEGG pathway analysis showed that the most significantly enriched pathway was the TNF signaling pathway and that, there are 62 mRNAs that might be associated with SCII among these co-expressed mRNAs.

## Prediction of cis-and trans-regulated genes of lncRNAs

Cis- and trans-regulated target genes of the DElncRNAs were predicted to further explore how lncRNAs regulate the pathological process of SCII. Thirty-four lncRNAs were found to have at least five cis-regulated target genes, some of which have been reported to be
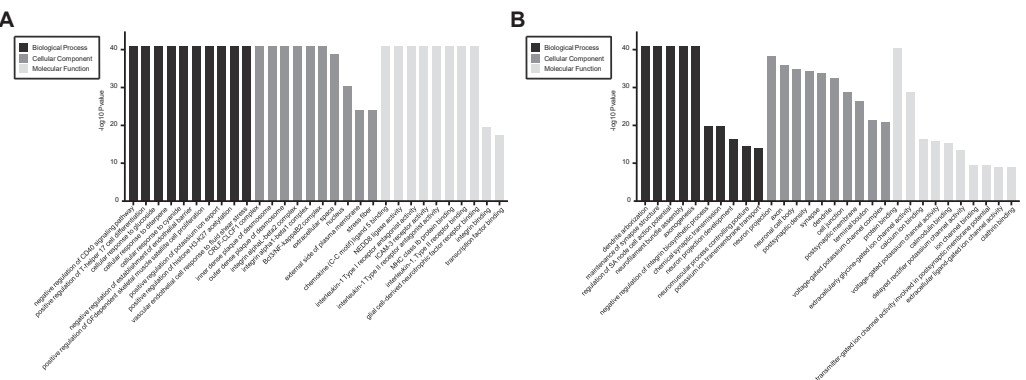

**Figure 4  GO enrichment analysis for the DEmRNAs with the 10 highest enrichment scores.** (A) GO enrichment analysis for up-regulated DEmRNAs; (B) GO enrichment analysis for down-regulated DEm-RNAs; red bars are biological processes, green bars are cellular components, and blue bars are molecular functions. The ordinate is the −Log10 *P*-value (−LgP). Larger −LgP values correlate with smaller *p*-values, indicating that the enrichment of differentially expressed genes in a given pathway is significant.

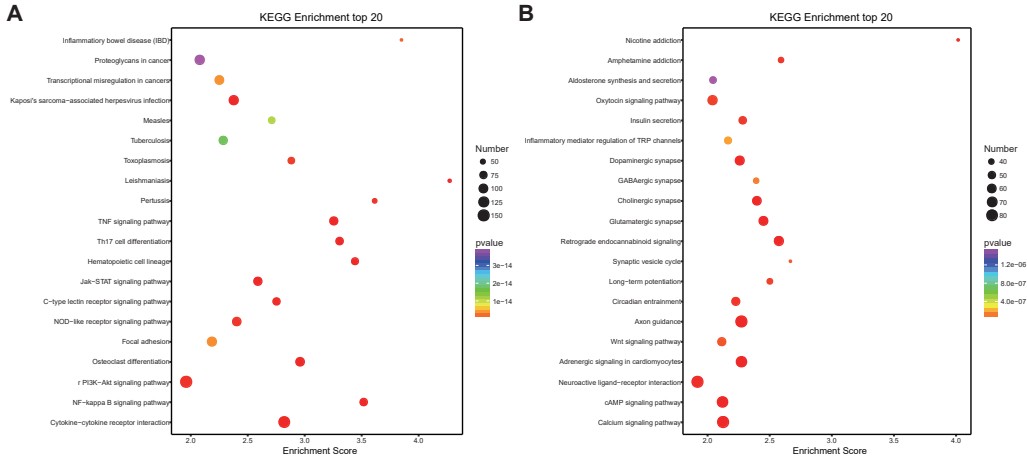

**Figure 5  KEGG pathway enrichment analysis of the DEmRNAs with the 20 highest enrichment scores.** (A) KEGG pathway enrichment analysis for up-regulated DEmRNAs; (B) KEGG pathway enrichment analysis for down-regulated DEmRNAs. The abscissa is the enrichment score. Size represents the number of enriched genes, and color indicates the degree of enrichment. Higher enrichment scores correlate with lower *P*-values, indicating that the enrichment of differentially expressed genes in a given pathway is significant.

associated with SCII (Fig. 8A). The top five lncRNAs ranked by the number of predicted nearby coding genes were TCONS_00026300, TCONS_00026304, TCONS_00026343, TCONS_00026342 and TCONS_00026425. LncRNAs and their trans-regulated genes are displayed in Fig. 8B. Each lncRNA has a different number of potential coding genes. For example, NONRATT007903.2 had a maximum of 12 trans-regulated genes, but NONRATT028952.2 and NONRATT001746.2 only had 1 trans-regulated gene.

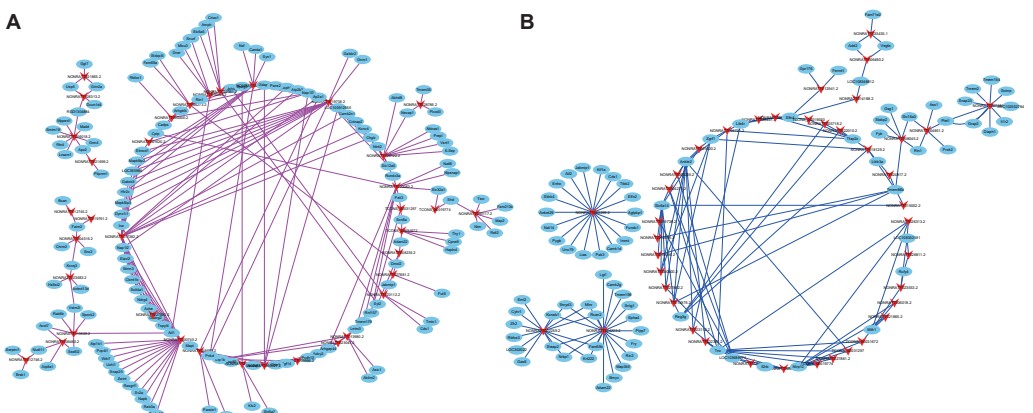

**Figure 6** **Co-expression network of the DElncRNAs and DEmRNAs.** (A) Positive co-expression network and (B) negative co-expression network of DElncRNAs and DEmRNAs. Arrows represent DElncRNAs, and ellipse nodes represent DEmRNAs. Red lines indicate positive co-expression relationships while blue lines indicate negative co-expression relationships between DElncRNAs and DEmRNAs.

## DISCUSSION

In the present study, we identified global expression changes of lncRNAs and possible relationships with coding genes in the delayed phase of SCII for the first time. A sum of 761 up-regulated and 694 down-regulated lncRNAs were found to be significantly differentially expressed 48 h in the rat spinal cord after SCII. Accordingly, 3,772 up-regulated and 2,935 down-regulated mRNAs were identified in the SCII model, suggesting that they were likely to be involved in SCII pathological processes.

Furthermore, we found that DElncRNAs and DEmRNAs were distributed on all chromosomes, indicating multiple potential functions they might play in SCII. Eight lncRNAs and 8 mRNAs were chosen for qRT-PCR, and the qRT-PCR results were in line with high throughput sequencing. Among them, Aqp4 (NM_001142366.2) was reported to be associated with a protective role of HMGB1 on astrocytic swelling after oxygen-glucose deprivation (*Sun et al., 2017*). Study also showed that Ryr2 (NM_001191043.1) in astrocytes and axons was increased after hypoxic injury, suggesting that it might have a neuroprotective role by reducing the cellular oxidative load (*Kesherwani & Agrawal, 2012*).

GO and KEGG pathway enrichment analysis revealed the role of differentially expressed lncRNAs. It is well established that the NF-$\kappa$B signaling pathway is closely associated with neuroinflammation and neuronal apoptosis in SCII (*Sun et al., 2017*; *Wang et al., 2017*). Induction of the cAMP signaling pathway is involved in improvement of neurologic function following SCII in mice (*Mares et al., 2015*). The TNF and Calcium signaling pathways are also speculated to be associated with SCII, but the precise roles of these pathways in SCII needs further research.

It is not easy to predict the lncRNAs function according to their nucleotide sequences due to a lack of primary sequence conservation leading to secondary structures (*Huang et al., 2011*). Thus, to further reveal the biological functions of DElncRNAs in SCII, lncRNAs/mRNAs co-expression networks were constructed. In all, 385 DEmRNAs and
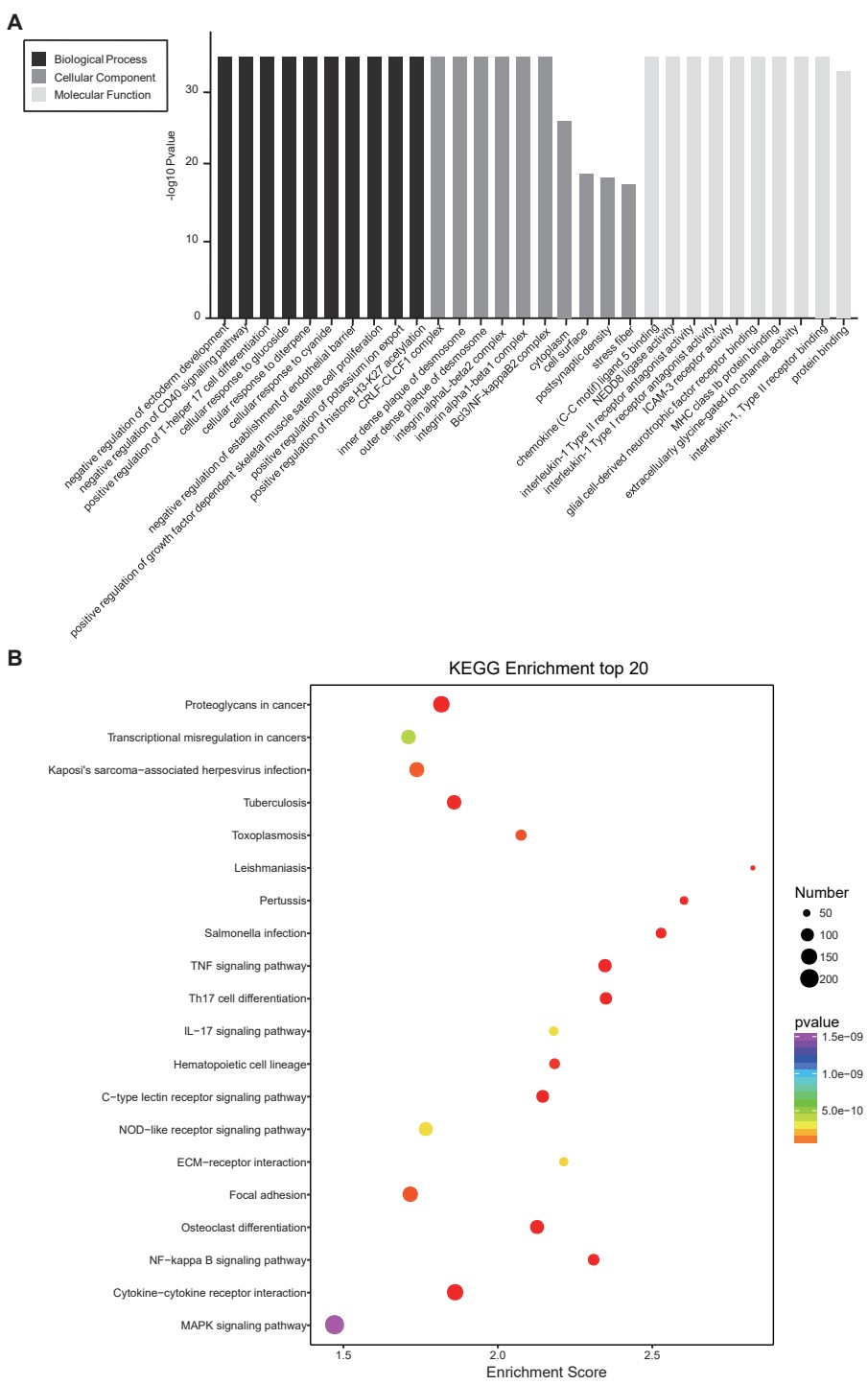

**Figure 7** **Functional annotation for co-expressed DEmRNAs of DElncRNAs.** (A) GO enrichment analysis for co-expressed DEmRNAs of DElncRNAs. Red bars are biological processes, green bars are cellular components, and blue bars are molecular functions. (B) KEGG pathway enrichment analysis for co-expressed DEmRNAs of DElncRNAs. The size of the spot indicates the gene numbers enriched in the pathway, and the color of the spot indicates the significance level of the enriched pathway.

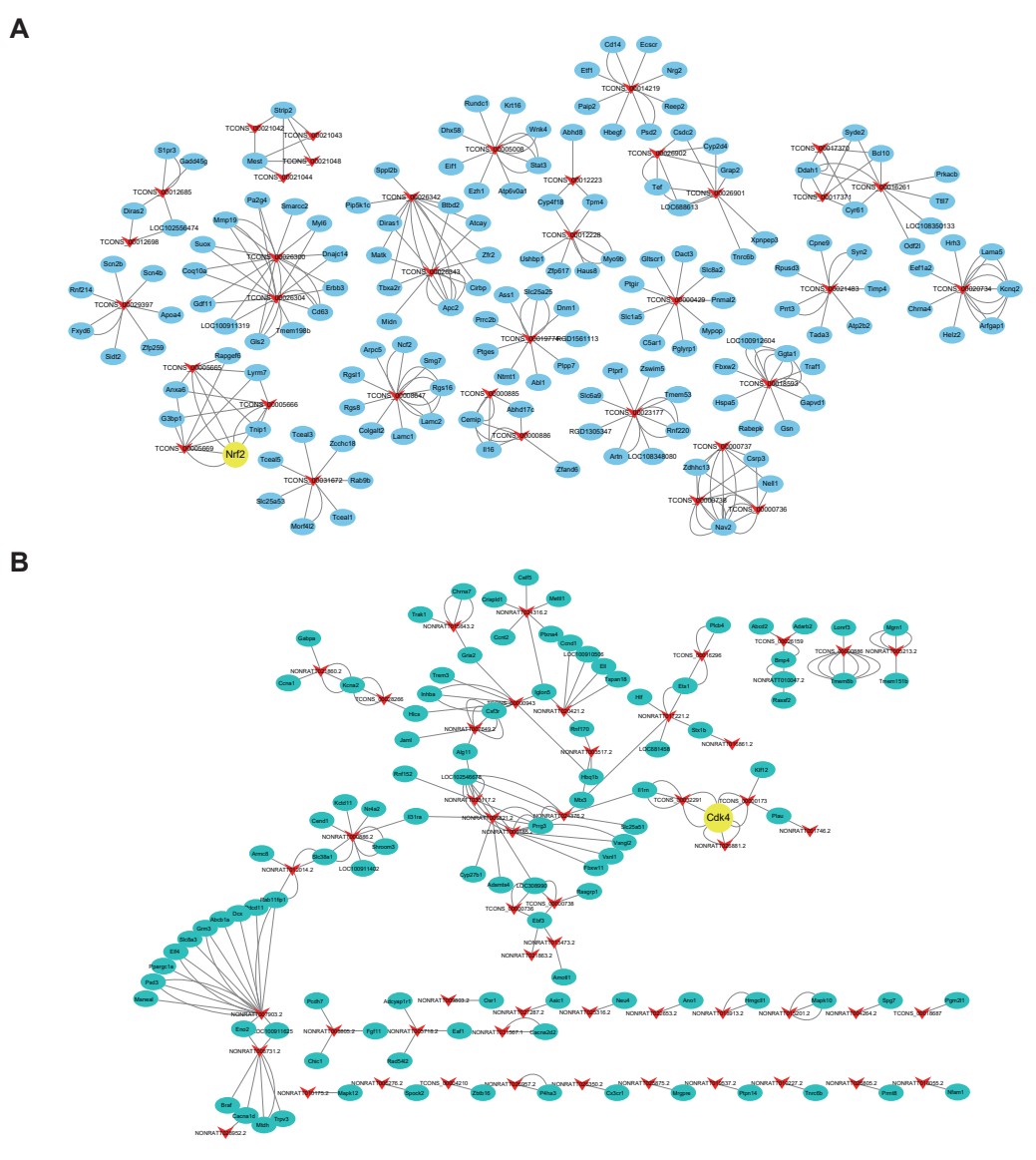

**Figure 8** **Construction networks of the DElncRNAs and their cis- and trans-regulated DEmRNAs.** (A) lncRNAs and their potential trans-regulated genes; (B) lncRNAs and their potential cis-regulated nearby genes. Arrows represent DElncRNAs, while the ellipse nodes represent trans- or cis-regulated DEmRNAs. Nrf2 and Cdk4 are marked in yellow.

221 DElncRNAs were identified in the network. Based on this network we described two different co-expression patterns (positive and negative co-expression) related to SCII, which suggested distinct underlying regulation mechanisms. Further functional annotation of the DEmRNAs co-expressed with the DElncRNAs indicated that several common and meaningful pathways were enriched, including TNF, NF- $\kappa$B and MAPK signaling pathways. The most significantly enriched pathway was the TNF signaling pathway, involving 62 co-expressed mRNAs. The TNF signaling pathway plays important roles in a variety of physiological and pathological processes, such as apoptosis, proliferation,

differentiation, immune response modulation and inflammation induction (*Bradley, 2008*; *McCoy & Tansey, 2008*). Studies have shown that ischemia reperfusion induced protein breakdown, lipid peroxidation and DNA damage, which lead to neuronal apoptosis and death, were primitively caused by amplifying the proinflammatory response (*Liu et al., 2017*; *Xie et al., 2017*). These data suggested that the TNF signaling pathway and associated DElncRNAs might be promising targets for further investigation.

Finally, we predicted corresponding mRNAs of DElncRNAs for functional annotation through cis- and trans-targeting. Cis-regulation is identified as lncRNA transcription affecting expression of neighboring genes (*Mercer, Dinger & Mattick, 2009*). LncRNA TCONS_00026300 and TCONS_00026304 were predicted to act on Rnf2 through cis-targeting. Nrf2 activation plays an important role in spinal cord ischemic tolerance (*Xu et al., 2014*). Nrf2 activation was involved in the therapeutic effect of methane on SCII via mediating anti-inflammatory and anti-apoptotic activities (*Wang et al., 2017*). In our study, the lncRNAs NONRATT026881.2, TCONS_00000173 and TCONS_00032291 were predicted to act on Cdk4 in a trans fashion. Cdk4 induction is involved in programmed cell death in SCII (*Sakurai et al., 2000*). Toghter, most DElncRNAs in the co-expression network have not yet been annotated, it is worth performing further studies to explore potential relationship between lncRNAs and SCII.

The major limitation of the present study is that we investigated lncRNA and mRNA changes 48 h after reperfusion, but did not detect their expression profiles at different time point or validate changes at the protein level. In addition, we did not select spinal cord areas for RNA isolation, RNA sequencing, or RT-PCR analyses. Besides that, the number of animals included in each group for RNA sequencing may be relatively small to identify DElncRNAs and DEmRNAs. Research with a time course and further experiments with more animal samples are needed to reveal roles of key lncRNAs and genes in SCII.

## CONCLUSIONS

In summary, differentially expressed mRNA and lncRNA transcripts involved in SCII were identified and validated using high-throughput RNA sequencing. Here, we indentified potential lncRNAs involed in the pathological process of SCII by four methods: 1. DElncRNAs selected with the criteria $p$-value <0.05 and |fold change|$\geq$ 2; 2. DElncRNAs involed in pathways predicted by KEGG pathway enrichment analysis; 3. DElncRNAs co-expressed with DEmRNAs; 4. DElncRNAs annotated through cis- and trans-targeting with DEmRNAs. These lncRNAs might be reliable candidates for further study by in vitro and in vivo experiments. Our results provided a basis for functional research on the lncRNAs involved in SCII. The results showed that specific lncRNAs may be important for diagnosis and therapy of SCII.

## ACKNOWLEDGEMENTS

We thank Yao Cheng for help in RNA-seq analysis.

### Funding

This study was financially supported by grants from the Shanghai Sailing Program (Grant No.17YF1425300). The funders had no role in study design, data collection and analysis, decision to publish, or preparation of the manuscript.

### Grant Disclosures

The following grant information was disclosed by the authors:
Shanghai Sailing Program: 17YF1425300.

### Competing Interests

The authors declare there are no competing interests.

### Author Contributions

- Zhibin Zhou analyzed the data, performed the experiments, prepared figures and/or tables, authored or reviewed drafts of the paper, and approved the final draft.
- Bin Han performed the experiments, prepared figures and/or tables, authored or reviewed drafts of the paper, and approved the final draft.
- Hai Jin authored or reviewed drafts of the paper, and approved the final draft.
- Aimin Chen conceived and designed the experiments, authored or reviewed drafts of the paper, and approved the final draft.
- Lei Zhu analyzed the data, conceived and designed the experiments, performed the experiments, authored or reviewed drafts of the paper, and approved the final draft.

### Animal Ethics

The following information was supplied relating to ethical approvals (i.e., approving body and any reference numbers):

All animal experiments were performed according to the guidelines of the Animal Ethics Committee of the Second Military Medical University (Shanghai, China).

### Microarray Data Deposition

The following information was supplied regarding the deposition of microarray data:

All new identified RNA sequencing are available at GEO: GSE138966.

### Data Availability

The raw measurements are available in the Supplemental Files.

### Supplemental Information

Supplemental information for this article can be found online at http://dx.doi.org/10.7717/peerj.8293#supplemental-information.

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
