# Peer review of "Changes in long non-coding RNA transcriptomic profiles after ischemia-reperfusion injury in rat spinal cord"

_PeerJ, doi:10.7717/peerj.8293_

## Round 0.1 · original submission · Major Revisions

I have now received all the reviews. As you can see, the reviewers ask, among others, for further clarifications and explanations on the analysis pipeline as well as some improvements on the used images.

The decision was 'Major revision', as I feel that following the reviewer's comments will help to improve the manuscript greatly.

I agree with the second reviewer, that the transparency and reproducibility of the used methods are of utter importance.

·

Basic reporting

1. English is used appropriately.
2. Literature references and background are provided sufficiently.
3. The quality of several figures should be greatly improved, such as Fiugure 4 and Figure 5.
4. The results positively support the scientific hypotheses. More lncRNA should be validated using qPCR.

Experimental design

The experimental desigh is reasonable.

Validity of the findings

More lncRNAs and mRNAs should be validated using qPCR.

Additional comments

This author Zhibin Zhou et al have provided the lncRNA and mRNA expression profiles after SCII and validated the differential expression lncRNA and mRNA. They further used bioinformatics tools to analyze the DElncRNAs and DEmRNAs. Generally, the article have some referenced value for developing novel diagnostic or therapeutic biomarkers for SCII. However, there are still some issues to be solved before publication.

Major revision
1. In Materials & Methods section, the number of the rats that were assigned into SCII group and sham group should be noted. What’s the L4 and L6 mean?
2. The primer sequences of the lncRNA and mRNA that were validated in qPCR assay should be provided.
3. The quality of several figures should be greatly improved, such as GO and KEGG figures.
4. More lncRNAs and mRNAs should be validated using qPCR.

Reviewer 2 ·

Basic reporting

Zhou et al. screened genome-wide expression patterns of lncRNAs and mRNAs in spinal cords from three SCII groups and three control tissues of the sham-operated group rats via RNA sequencing. They did gene ontology (GO) and KEGG pathway analyses as well as built co-expression networks to explore the functions of DElncRNAs and DEmRNAs.
RNA-seq is a powerful tool that has been used to identify novel protein-coding and non-coding RNA transcripts involved in the regulation of gene expression. Noncoding RNAs (ncRNAs) comprise a majority of the genome and play important developmental and physiological roles, as well as roles in disease mainly by regulating gene expression. Long noncoding RNAs (lncRNAs) are traditionally defined as having greater than 200 nucleotides. Accumulating evidence has shown that lncRNAs have various biological functions in RNA processing and translation and act as important gene expression regulators, however, few studies have explored the important role of the neuroprotective effect LncRNA of the spinal cord during ischemia-reperfusion injury.
Zhou et al findings could provide basic functional research on the lncRNAs involved in SCII and might be critical for providing new insights into the identification of potential targets for SCII therapy.

I have several major comments about this work as follows:

1. Your introduction needs more detail. I suggest that you provide more justification for your study and mention other similar studies. Specifically, you should expand upon the knowledge gap being filled. For example, there is similar study “https://www.ncbi.nlm.nih.gov/pmc/articles/PMC6488167/” and “https://www.ncbi.nlm.nih.gov/pubmed/29998804“ I suggest the authors compare the result with previous similar studies.
2. The full bioformatics pipeline, the commands used and the parameters need to be provided. I would like to be able to see and verify the main analysis of the paper.
3. Also, the raw data should be uploaded in a public repository and a link provided in the paper
4. The authors need to be much more clear in the exact steps taken to do the statistical calculation. For example, the approach used for the prediction of cis- and trans- regulated genes of LncRNAs is not written. It should be written in detail.
5. The authors claimed to have seen 1455 DELncRNAs and 6707 DEmRNAs genes, while previous studies had a much smaller number of DE genes. How do the authors explain this discrepancy?
6. The MDS or PCA of the samples are needed for both LncRNAs and mRNA.
7. I would like to see additional figures for the figures 5A, 5B and 7B to have the over-represented p-value adjustment instead of p-value and enrichment score.

Experimental design

I have several major comments about this work as follows:
1. Various cytokines peak in the serum within the first 24 hr of ischemia-reperfusion injury. It needs justification of why the authors decided to study 48 hours after the reperfusion to harvest L4 to L6 spinal cords?
2. The number of samples (n=6) is not enough for making a strong claim. I do understand that it might be not feasible to have additional samples, authors at least need to discuss this issue in the paper.
3. SCII rat models have a bigger spinal cord infarct zone than the control rat spinal cord. I was wondering if the authors provide this factor during their experiment.
4. Which neurological function assessment they have done during their experiment

Validity of the findings

no comment

Additional comments

no comment

Reviewer 3 ·

Basic reporting

I think the figures were not of sufficient resolution, which should be improved.

Experimental design

no comment

Validity of the findings

no comment

Additional comments

Zhou et al investigated the differentially expressed lncRNAs (DElncRNAs) and mRNAs (DEmRNAs) in rat spinal cords in the SCII model by high-throughput RNA sequencing method. They found that SCII could reshape the expression patterns of LncRNAs and mRNAs in the rat spinal cord after SCII. These results provide a lot of potential molecules that may play some role in post-SCII pathophysiological processes. This is a well-performed study with correct bioinformatics approaches and correct statistical analysis. I only have some minor comments that need to be considered by the authors before accepted for publication.

1 Although English is of a very good standard, some minor errors remain. A thorough check should be performed by the authors
2 Clear, and Consistent font type and size should be used in the author's figures.
3 Figure 1. C and D should have the same size.
3 The resolution of Figures 4, 5 and 6 should be improved.

---

## Round 0.2 · Minor Revisions

As you can see from the attached reviewer reports, one reviewer is still requesting a few minor changes to the manuscript, namely to clarify some contradictions as well as small clarifications of the results to further improve your submitted manuscript that need to be addressed prior to acceptance of the manuscript.

·

Basic reporting

no comment

Experimental design

no comment

Validity of the findings

no comment

Additional comments

The author has revised the manuscript according to my comments. In my opinions, the manuscript could be accepted just before some minor revisons are made.
1. The author forgot to describe the validation results of DElncRNA and DEmRNA by qPCR. Please add the corresponding content.
2. The criteria for screening the DElncRNA and DEmRNA is contradictory in Abstract, Materials & Methods and Results. FC ≥1.5 or FC ≥2.0? Why the author choose this criteria for screening the DElncRNA?
3. In the discussion, it is better to provide more directions about how to study the DElncRNAs in this research in the future. The author could tell us which lncRNA is the potential diagnostic or therapeutic target for ischemia-reperfusion injury.

Reviewer 2 ·

Basic reporting

The authors have adequately responded to my comments. I have no further comments.

Experimental design

No further comments.

Validity of the findings

No further comments.

Additional comments

No further comments.

Reviewer 3 ·

Basic reporting

no comment

Experimental design

no comment

Validity of the findings

no comment

Additional comments

The authors have addressed all of the criticisms. I have no additional comments and suggest acceptance.

---

## Round 0.3 · accepted · Accept

Since the previous decision was 'Minor revision' and I can see that you addressed all raised points, I feel that it is not required anymore to send the manuscript to the reviewers for another round of reviewing.